# Parametric Studies on Finishing of AZ31B Magnesium Alloy with Al_2_O_3_ Magnetic Abrasives Prepared by Combining Plasma Molten Metal Powder with Sprayed Abrasive Powder

**DOI:** 10.3390/mi13091369

**Published:** 2022-08-23

**Authors:** Zhihao Li, Yugang Zhao, Guangxin Liu, Chen Cao, Qian Liu, Dandan Zhao, Xiajunyu Zhang, Chuang Zhao, Hanlin Yu

**Affiliations:** School of Mechanical Engineering, Shandong University of Technology, Zibo 255049, China

**Keywords:** magnetic abrasive finishing, Al_2_O_3_ magnetic abrasive, AZ31B magnesium alloy, RSM

## Abstract

High-performance iron-based Al_2_O_3_ magnetic abrasive powder (MAP) prepared by combining plasma molten metal powder with sprayed abrasive powder is used for magnetic abrasive finishing (MAF) of AZ31B magnesium alloy to remove surface defects such as creases, cracks, scratches, and pits generated during the manufacturing process of the workpiece, and to reduce surface roughness and improve its wear and corrosion resistance. In order to solve the problem of magnetic abrasive powder splash in the MAF process, the force analysis of the MAP in the processing area is conducted, and a composite magnetic pole processing device was designed and simulated to compare the effects of both devices on MAF, confirming the feasibility of composite magnetic pole grinding. Then, experiments have been designed using Response Surface Methodology (RSM) to investigate the effect of four factors-magnetic pole rotation speed, grinding gap, magnetic pole feed rate, magnetic abrasive filling quantity-on surface roughness and the interactions between them. The minimum surface roughness value that can be obtained is used as the index for parameter optimization, and the optimized parameters are used for experiments, and the results show that the established surface roughness model has good predictive ability.

## 1. Introduction

As one of the lightest metal structural materials, magnesium alloy has the advantages of good shock absorption, low density, high specific strength, easy recycling, resistance to electromagnetic interference, etc. It is widely used in aerospace, electronic engineering, transportation and 3C products [1,2,3]. Because its elastic modulus is similar to that of human bone and it has excellent biocompatibility, magnesium alloy has become a common and popular material in the field of biomedical implantable materials [4,5,6]. However, due to its poor wear and corrosion resistance in the human body environment, resulting in a rapid degradation rate that may cause the implant to fail before the bone is fully healed, it is significant to reduce the surface roughness and improve the surface integrity of the magnesium alloy, thereby improving its wear and corrosion resistance [7,8,9,10,11]. Polishing the magnesium implants to improve their surface quality also facilitates subsequent thin film plating and drug coating [11]. However, due to the flammability and fragility of magnesium alloy, it is hard to polish its surface by traditional processing methods [12,13]. Therefore, this study proposes to process magnesium alloy by magnetic abrasive finishing.

Magnetic abrasive finishing (MAF) is an ultra-precise surface special processing technology, with the advantages of small temperature rise, high self-adaptability, easy to achieve automation, and no tool compensation [14,15]. MAF is widely used for flat surface finishing of various metal and non-metal materials, and has achieved good results. Wu et al. used a low-frequency alternating magnetic field for magnetic grinding finishing of SUS304 stainless steel and investigated the effects of magnetic field distribution, grinding fluid, magnetic pole speed, and current frequency on surface roughness and material removal rate [16]. Zhang et al. used magnetic particle grinding technique to polish 316 L stainless steel samples generated by selective laser melting (SLM) and designed an analytical model for material removal rate by correlating the surface roughness of the workpiece with the indentation depth [17]. Khalil designed a magnetic field assisted machining system (MFAM) and combined it with single point diamond cutting (SPDT) to improve the machinability and surface quality of Ti6Al4V alloy parts [18]. However, there are few studies on the magnetic abrasive finishing of medical magnesium alloys.

The properties of magnetic abrasive powders (MAPs) have a crucial influence on the processing quality of magnetic abrasive finishing [19]. Currently, the common methods of MAP preparation in the market are mechanical mixing method, bonding method, sintering method, and chemical compound plating method [20,21]. However, MAPs prepared by the above methods suffer from problems such as low efficiency due to abrasives fly-off, poor abrasive morphology, poor grinding performance, and short service life. Moreover, the preparation devices are complicated and costly. Combination plasma molten metal powder and sprayed abrasive powder is a novel preparation method of high performance magnetic abrasive powders, which is economical and efficient and is expected to be industrialized [22].

In this study, a composite magnetic pole device is proposed to finish the surface of AZ31B magnesium alloy. The MAPs were selected as Al_2_O_3_ magnetic abrasives prepared by combining plasma molten metal powder with sprayed abrasive powder. The effect of magnetic pole rotation speed and magnetic induction strength on the MAF process was obtained by force analysis of MAP. Subsequently, the magnetic induction intensity simulation and analysis were carried out for the two processing methods of single magnetic pole and composite magnetic poles. Finally, by response surface experimental design method, the response surface analysis model of magnetic pole rotation speed, grinding gap, magnetic pole feed rate and magnetic abrasive filling quantity on surface roughness was established to analyze the influence law of process parameters on surface roughness and the interaction between process parameters, and to obtain the best combination of process parameters on surface roughness.

## 2. Magnetic Abrasive Finishing Mechanism

### 2.1. Principle of Planar Magnetic Abrasive Finishing Process

Magnetic abrasive finishing is a flexible ultra-precision machining technology, in which the magnetic abrasive powder is pressed onto the surface of the workpiece by the force of a magnetic field generated by a permanent magnet or electromagnet, resulting in a surface micro-cutting finishing process. Figure 1 shows the principle of planar MAF with auxiliary magnetic pole. Clamp the magnetic source device on the milling machine spindle and leave a certain grinding gap between the magnetic pole and the workpiece for the filling of the magnetic abrasive powders. The auxiliary pole is mounted below the workpiece to enhance the magnetic field strength in the machining area. The MAPs consist of a hard abrasive phase and a magnetic iron-based phase. The ferromagnetism of the iron matrix makes the MAPs align along magnetic susceptibility lines to form a “Flexible Magnetic Abrasive Brush (FMAB)”. This FMAB is tightly attached to the surface of the workpiece, with a certain degree of rigidity, will not easily deformation, and its inlaid hard abrasive phase hardness is greater than the workpiece hardness, so under the action of the magnetic field force will produce a certain indentation depth on the surface of the workpiece. At the same time, the spindle of the milling machine drives the magnetic pole to rotate and can reciprocate, the FMAB on the pole has relative motion with the workpiece, and the hard abrasive particles remove the surface material of the workpiece, thus realizing the finishing process and improving the surface quality of the workpiece.

### 2.2. Assumptions of the Model

Force analysis of MAP in the MAF process is a complex problem. To simplify it, the following assumptions are made:(1)The shape of prepared MAPs is almost spherical, so it can be assumed that the model of MAPs is perfect spherical.(2)The Al_2_O_3_ hard particles on the MAPs are uniformly distributed and the shape of the cutting edge is considered spherical. The MAP maintains its properties and shape throughout the MAF process.(3)The MAF process is stable and the FMAB is continuous and uniform.

### 2.3. Force Analysis of MAP

During the MAF process, the MAPs are attracted by the magnetic field to form a flexible magnetic abrasive brush and follow the magnetic pole. The motion and force of the flexible magnetic brush is similar to that of a single abrasive in the grinding area, so the motion of the entire flexible magnetic brush can be approximated by the force analysis of a single abrasive. Figure 2 shows the force analysis of a single MAP in the grinding area. MAP is subject to two kinds of force, which are the magnetic field force generated by the magnetic poles on the abrasive particles and the dynamic force generated by the circular motion of the class. Where the magnetic field force provides the force for MAP adsorption and the normal force is obtained by adjusting the grinding gap. The magnetic field force is divided into the force along the direction of magnetic susceptibility and the force along the direction of isomagnetic potential, calculated as [23]
(1)$$F_{x}=V_{m}x_{m}H\frac{\partial H}{\partial x}$$
(2)$$F_{y}=V_{m}x_{m}H\frac{\partial H}{\partial y}$$
where, $V_{m}$ (µm^3^) is the volume of a single MAP, $x_{m}$ is the magnetization rate of MAP, H is the magnetic field strength at the location of MAP, $\frac{\partial H}{\partial x}$, $\frac{\partial H}{\partial y}$ are the gradients of the magnetic field intensity variation along the magnetic lines of force and isotropic directions, respectively. The magnetic field force $F_{m}$ is the combined force of $F_{x}$ and $F_{y}$, calculated as
(3)$$F_{m}=\sqrt{F_{x}^{2}+F_{y}^{2}}$$

The magnetic induction strength is calculated as
(4)$$B=\mu_{0}(H+M)$$
where, $\mu_{0}$ is the vacuum magnetic permeability. Combining the above equations gives the magnetic field force $F_{m}$ as
(5)$$F_{m}=x_{m}H\sqrt{{\left(\frac{\partial H}{\partial x}\right)}^{2}+{\left(\frac{\partial H}{\partial y}\right)}^{2}}$$

Therefore, when the grinding gap is certain, the higher the magnetic field strength, the higher the magnetic field force on the abrasive grains. The greater the grinding pressure obtained by adjusting the grinding gap, the deeper the indentation depth produced by MAPs on the workpiece surface, and the better the grinding effect. Therefore, the combination of process parameters in the magnetic grain finishing process can be adjusted to change the abrasive pressure of the magnetic abrasive on the workpiece surface to achieve efficient material removal. The normal force of MAP $F_{n}$ obtained by adjusting the process parameters is calculated as
(6)$$F_{n}=\frac{B^{2}}{16\mu_{0}}\cdot \frac{3\pi^{2}d_{m}^{2}V_{m}(\mu_{i}-1)}{3V_{m}(2+\mu_{i})+\pi V_{i}(\mu_{i}-1)}$$
where, $d_{m}$ (µm) is the diameter of MAP, $\mu_{i}$ is the relative permeability of magnetic abrasives, $V_{i}$ (µm^3^) is the volume of iron matrix in a single MAP.

The MAP is subjected to a magnetic force $F_{m}$, which has an angle α with the horizontal plane and a projection on the horizontal plane with an angle β with the *X*-axis. At the beginning, the angle α=0, the component of the magnetic field force on the MAP in the vertical direction is used to balance its own gravity. Due to the action of the normal force $F_{n}$ magnetic abrasives are pressed into the surface of the workpiece to form an indentation depth h. During the MAF process, the MAP pressed into the workpiece surface is subject to a small hysteresis by the cutting resistance $F_{f}$. At this time, the angle α>0, and the component force of $F_{m}$ in the *Y*-axis direction provides the cutting force $F_{t}$ for the magnetic abrasive finishing process, calculated as
(7)$$F_{t}=F_{m}\cdot \sin\alpha \cdot \cos\beta$$

The dynamic force generated by the magnetic abrasive doing a class of circular motion is the centrifugal force $F_{c}$ away from the center of the circle, and the magnitude is calculated as
(8)$$F_{c}=4\pi^{2}mrn^{2}$$
where, m (kg) is the mass of MAP, r (m) is the radius of rotation of the MAP, n (r·min^−1^) is the magnetic pole rotation speed.

Thus, the higher the magnetic pole rotation speed, the greater the centrifugal force on the abrasive. As shown in Figure 3, during the MAF process, depending on the magnetic pole rotation speed, there are three types of relationship between the centrifugal force applied to the abrasive and the magnitude of the division of the magnetic adsorption force in this direction, corresponding to the three states of the magnetic brush. When the magnetic pole rotation speed is too high, the magnetic abrasives will leave the grinding area under the action of centrifugal force and gather or fly away at the edge position of the magnetic pole, resulting in lower grinding efficiency. When the magnetic pole rotation speed is moderate, the formed magnetic brushes are neatly and evenly arranged along the radial direction of the main magnetic pole. When the speed is too low, the magnetic abrasives will gather at one point, and the interaction between the MAPs will lead to the damage of the cutting edge of the abrasive grain and shorten the service life. In order to maintain a good cutting performance of the magnetic brush, the force situation should be satisfied as follows
(9)$$F_{c}\leq F_{m}\cdot \cos\alpha \cdot \cos\beta$$

### 2.4. Analysis of Magnetic Field

The material removal in the magnetic abrasive finishing process is mainly done by pressing the hard abrasive phase of the MAP into the surface of the workpiece by normal force *F_n_*, which is then driven by the cutting force in the tangential direction. When the shape of the abrasive phase pressed into the surface of the workpiece and the angle of contact with the hill of surface roughness are suitable, the workpiece material will be removed. The indentation depth is calculated as [24]
(10)$$h=\frac{d_{m}}{2}-\sqrt{\frac{d_{m}^{2}}{4}-\frac{F_{n}d_{m}^{2}}{4KH_{w}}}$$
where, $H_{w}$ (HB) is the Brinell hardness of the workpiece material, K is a factor determined by the type of material [25].

It can be seen that the indentation depth of magnetic abrasives in MAF processing increases with an increasing grinding pressure, which is related to the magnetic field strength, so the grinding pressure can be increased by increasing the magnetic field strength to make the magnetic abrasives completely driven by the magnetic poles, so that the surface of the workpiece can be sufficiently ground and processed. In addition, improving the magnetic field distribution and making the magnetic induction intensity in the grinding area more uniform can improve the binding of the flexible magnetic brushes, which in turn improves the grinding efficiency.

Magnetic abrasive finishing is a free abrasive processing method, and the variation of the magnetic field gradient has a great influence on the quality of MAF processing [26]. The magnetic field gradient is the rate of change of the magnetic field strength with spatial displacement, denoted by the symbol dH/dX. It is a vector quantity that points in the direction of the largest change in the magnetic field gradient. During the MAF process, MAPs are in free state and are accompanied by flow and mutual extrusion. When the magnetic abrasive at the bottom of the FMAB is unable to cut the peak surface roughness due to hard abrasive wear or wrong cutting angle, it will rotate under the action of magnetic field gradient and flow from the area with relatively weak magnetic induction intensity to the area with relatively strong magnetic induction intensity, thus completing the renewal of MAP. If the magnetic field gradient does not change significantly, the abrasive at the bottom of the magnetic brush will be worn out due to untimely renewal, thus resulting in lower grinding efficiency and low abrasive utilization. It has been shown that slotting the magnetic pole surface and increasing the magnetic induction strength are beneficial to improve the magnetic field distribution and the variation of the magnetic field gradient. Therefore, in this study, we improved the magnetic induction during the MAF process by adding an auxiliary pole under the workpiece. This improves the grinding efficiency. As for the auxiliary pole, it is actually another piece of permanent magnet.

In order to visually analyze the distribution and strength of magnetic field during the MAF process, the grinding gap between the main pole and the workpiece surface is set to 1.5 mm, and the magnetic field simulation is carried out separately for the two processing models by Comsol software. The main magnetic pole size is Ø29 mm × 8 mm and has a through hole of Ø8 mm in the center, and the size of the auxiliary pole immediately below the workpiece is 29 mm × 20 mm × 5 mm. The material of both poles is set to Nd-Fe-B, and the magnetization direction is set to the negative direction of *Y*-axis. The workpiece material is set to magnesium AZ31B, size is 29 mm × 20 mm × 5 mm, and the rest of the parameters are determined by the software according to the set materials. Figure 4 shows the simulation diagram of magnetic induction intensity of two MAF methods. As shown in Figure 4a, when no auxiliary pole is installed under the workpiece, only the magnetic field gradient near the pole edge changes more obviously in the whole grinding area, which causes the MAPs near the pole center to gather towards the pole edge. When the grinding gap is reduced, the interaction between these clustered magnetic abrasives becomes more pronounced, resulting in an uneven distribution of normal force. The final performance is uneven grinding efficiency, the surface of the workpiecee appears a big gap, and even the phenomenon of plowing and over-grinding, which destroys the surface integrity of the workpiece. The simulation results of the composite magnetic pole are shown in Figure 4b. After installing the auxiliary magnetic pole under the workpiece, the magnetic field gradient of the whole grinding area changes in a clear trend, and the magnetic abrasive renewal is more active during MAF processing, which significantly improves the utilization rate of the abrasive. The magnetic field distribution is more uniform, which leads to the formation of neatly arranged and evenly spaced magnetic brushes, effectively improving the grinding efficiency and surface quality.

The magnetic induction intensity of a line taken on a symmetrical section of the workpiece simulation surface is plotted on Figure 5. As shown in Figure 5a, when the auxiliary pole is not installed, the maximum magnetic induction intensity of the workpiece surface grinding area is 552 mT, and the average value is 427 mT. In contrast, the maximum magnetic induction intensity of the workpiece surface grinding area of the composite pole is 583 mT, and the average value is 536 mT, and the average magnetic induction intensity is about 25% higher. In summary, the use of composite magnetic poles can improve the magnetic field distribution in the grinding area, while increasing the magnetic induction intensity, thus reducing the uneven grinding caused by magnetic abrasive clusters and further improving the surface quality of the workpiece.

## 3. Experiments

### 3.1. Experimental Materials

The MAP used in this test is Al_2_O_3_/Fe-based magnetic abrasive powders prepared by combining plasma molten metal powder with sprayed abrasive powder, which are high-performance bonded abrasive. Figure 6 [22] shows the micrographs of Al_2_O_3_/Fe-based magnetic abrasives under Scanning Electron Microscopy (SEM). The MAP is an ideal sphere, Al_2_O_3_ hard abrasive is evenly and densely distributed in the surface layer of the iron matrix, part of it is firmly embedded in the iron matrix, and the other part protrudes in the surface of the iron matrix and remains intact, with good micro cutting performance.

Figure 7 [22] shows the XRD graph and EDS surface scan of Al_2_O_3_ magnetic abrasive. X-ray diffractometer (D8 Advance, Bruker) is used to analyze the surface phase composition of Al_2_O_3_ MAPs. For all MAPs, two diffraction peaks of 44.673° and 82.333° are detected in the α-Fe phase. Three peaks of 43.173°, 57.170° and 68.118° are obtained in the Al_2_O_3_ phase. It means that only α-Fe and Al_2_O_3_ hard abrasive phases exist in the prepared magnetic abrasives, which ensures the soft magnetic properties and cutting performance of MAP. The EDS surface scan Figure 7b shows that there is no significant component diffusion between the elements of the Fe matrix phase and the elements of the abrasive phase at the interface, indicating that the type of bonding of the two phases is a strong mechanical bond formed by rapid cooling and solidification [22]. It means that MAPs can maintain good cutting performance during MAF process.

The machined workpiece used for the experiments are biomedical AZ31B magnesium alloy plate with dimensions of 29 mm × 20 mm × 5 mm. The chemical composition of AZ31B magnesium alloy is shown in Table 1 [27], and the performance indexes of AZ31B magnesium alloy are shown in Table 2 [13].

### 3.2. Experimental Setup

The experimental platform for this study is a modified XK7136C CNC milling machine, as shown in Figure 8. The tool tightening structure of the CNC milling machine was modified to make it possible to install a magnetic pole spindle system. A composite magnetic pole device is used, where the main pole is bolted to the milling machine spindle and the grinding gap between it and the workpiece surface is filled with a certain amount of Al_2_O_3_/Fe-based magnetic abrasive prepared by combining plasma molten metal powder with sprayed abrasive powder. The auxiliary pole is mounted under the workpiece, close to the lower surface of the workpiece, to provide stronger magnetic field strength for the MAF process. Both the main and auxiliary pole are made of Nd-Fe-B permanent magnets, which have the advantages of small size, light weight, no energy consumption and simple structure, and no eddy current heat is generated during the processing. The MAP is pressed into the surface of the workpiece by adjusting the grinding gap, and then rotates and feeds horizontally under the drive of the spindle, thus finishing the surface of the workpiece.

Before each group of test, the surface of AZ31B magnesium alloy plate was firstly pretreated with sandpaper, 5 different areas were evenly taken in the grinding area, and the surface roughness of these 5 areas were measured by a DSX1000 3D digital microscope (Olympus, Tokyo, Japan) and the average value was taken as the initial surface roughness before the magnetic finishing process. The DSX1000 3D digital microscope acquires images from the bottom up layer by layer and stitches them together into a high resolution 3D image. Besides, it can also give the required inspection results quickly and efficiently.

### 3.3. Response Surface Method Process Parameter Design

Response surface method (RSM) is a nonlinear modeling and analysis method that integrates optimization design and statistical analysis based on the optimization process of experimental conditions. By combining the scheme with the experiment, the response values corresponding to each group of parameters are obtained, and the response surface model between the variables and the response values is constructed so as to establish the functional relationship between the response target and the design variables, and subsequently the optimal process parameters are obtained by analyzing the functional relationship. The Box-Behnken response surface experimental design method in RSM was chosen for the experimental design and results analysis of the magnetic particle grinding process of AZ31B magnesium alloy material. The magnetic pole rotation speed, grinding gap, magnetic pole feed rate and magnetic abrasive filling quantity are taken as the main influencing factors, and a four-factor, three-level factor level table was designed for the experiment with surface roughness as the response value. The experiment parameters selected based on previous experience are: magnetic pole rotation speed *n* is 800~1600 r/min, grinding gap d is 1~3 mm, magnetic pole feed rate f is 1–3 mm/min, magnetic abrasive filling quantity m is 1.5~2.5 g, and the grinding time is 15 min. With the surface roughness as the response value, the above four factors were coded separately, and the levels are shown in Table 3.

### 3.4. Experiment Results and Discussion

The surface roughness of the machined AZ31B magnesium alloy plate was taken as the response index by the same method, and the test parameters and results are shown in Table 4.

The experiment data were fitted using Design-Expect software to obtain the regression equations between surface roughness Ra and four process parameters: magnetic pole rotation speed (*A*), grinding gap (*B*), magnetic pole feed rate (*C*), and magnetic abrasive filling quantity (*D*). The expression is as follows
(11)$$\begin{array}{c} Ra=0.13-0.051\times A+0.024\times B+0.023\times C-0.011\times D-0.0075\times AB-0.0075\times AC\\ +0.00975\times AD+0.00025\times BC+0.0085\times BD+0.00575\times CD+0.001333\times A^{2}\\ +0.008583\times B^{2}-0.002292\times C^{2}+0.003458\times D^{2} \end{array}$$

In order to be able to accurately predict the surface roughness of AZ31B magnesium alloy material after machining, the normal distribution of the residuals of the prediction model was established, as shown in Figure 9. The distribution of each test sample point on a straight line and on both sides of the line indicates that the model prediction and the actual value for surface roughness can match well, and the preliminary judgment can be used to predict the surface roughness after finishing.

Table 5 shows the ANOVA results of the surface roughness regression model. The F-value represents the significance of the whole regression equation model and *p* represents the significance level of the regression equation model. The number of independent variables is 14 and the degree of freedom is 12. Checking the F-test (F-test) critical value table, the standard F-value for the given significance level at the significant level α = 0.05 is 2.534, and the F-value of the regression model of surface roughness is 22.53 > F_0.05_ (14,12) = 2.424, indicating that the prediction model of the regression equation established between surface roughness and the four independent variables was highly significant, indicating a high degree of confidence in this prediction model. The misfit *p* value is 0.0668 > 0.05, which is not significant relative to the pure error. Analysis of the *p* values and F values corresponding to the four factors in the table shows that *p*(A), *p*(B) and *p*(C) are less than 0.0001, indicating that the magnetic pole rotation speed, grinding gap and feed rate are highly significant with the surface roughness Ra. *p*(D) was less than 0.05, indicating that the magnetic abrasive filling quantity was significantly related to Ra. The values of F show F(A) > F(B) > F(C) > F(D) in descending order, from which it can be concluded that the parameters that have the greatest influence on Ra are the magnetic pole rotation speed, grinding gap, magnetic pole feed rate and magnetic abrasive filling quantity in descending order. The multivariate correlation coefficient R-Squared value is 0.9634 and the corrected multivariate correlation coefficient Adj R-Squared value is 0.9206, which indicates that the prediction model can explain 92.06% of the response values and has a high predictive power.

### 3.5. Response Surface Interaction Analysis

According to the prediction equation of Ra, after determining the level of certain influencing factors, the influence law of the interaction between the remaining two factors on Ra can be obtained. Using the zero level of the process parameters as a reference, the interaction between the factors is discussed in terms of their respective effects on Ra.

#### 3.5.1. The Interaction of Magnetic Pole Rotation Speed and Feed Rate

The Figure 10 shows the 3D surface map and contour map of the interaction between magnetic pole rotation speed and feed rate on surface roughness. It can be seen that the surface roughness value of the workpiece tends to increase as the magnetic pole speed decreases and the feed rate increases when the grinding gap and the magnetic abrasive filling quantity are fixed at the middle value. While a large pole rotation speed and a small grinding gap are necessary to obtain a small surface roughness. This is because the magnetic abrasives are magnetized to form a magnetic brush, which is attracted to the magnetic pole surface and rotates and feeds horizontally with the pole to form a spiral line of abrasive marks on the surface of the workpiece. The faster magnetic pole rotation speed and slower feed rate form denser abrasive traces, which makes the magnetic abrasive and the workpiece surface have more relative movements, and the material removal increases, which is conducive to the removal of the surface defect layer.

#### 3.5.2. Interaction of Grinding Gap and Magnetic Abrasive Filling Quantity

The Figure 11 shows the 3D surface map and contour map of the interaction between grinding gap and magnetic abrasive filling quantity on surface roughness. It can be seen that, when the magnetic pole rotation speed and feed rate are fixed at intermediate values, a larger grinding gap and a smaller abrasive filling quantity do not result in lower surface roughness values. This is because the magnetic induction intensity in the machining area is influenced by the grinding gap, and the grinding pressure on the magnetic abrasive is proportional to the magnetic induction intensity according to the formula. When the grinding gap increases, the magnetic induction intensity reflected by the magnetic poles to the surface of the workpiece will be reduced, and the grinding pressure on the abrasive will be reduced, resulting in a reduction of the indentation depth of the abrasive pressed into the surface of the workpiece, and the original burrs, scratches and other defects on the surface of the workpiece will not be completely removed, and a better surface quality cannot be achieved. The amount of magnetic abrasive filling quantity has a great influence on the quantity and quality of the magnetic brushes formed between the magnetic poles and the surface of the workpiece. When the grinding gap is small and the magnetic abrasive filling quantity is large, the magnetic abrasive forms a dense and uniform magnetic brush in the grinding gap, the relative contact area between the abrasive and the workpiece surface increases, and the surface roughness value of the machined workpiece decreases.

#### 3.5.3. Interaction of Magnetic Pole Rotation Speed and Magnetic Abrasive Filling Quantity

The Figure 12 shows the 3D surface map and contour map of the interaction between magnetic pole rotation speed and magnetic abrasive filling quantity on surface roughness. It can be seen that smaller magnetic pole rotation speeds and smaller magnetic abrasive filling quantity do not result in lower surface roughness values when the grinding gap and magnetic pole feed rate are fixed at intermediate values. The larger the magnetic pole rotation speed, the smaller the effect of magnetic abrasive filling quantity on the surface roughness value. When the magnetic pole rotation speed is small, the magnetic abrasives will gather at the point of higher magnetic field strength in the processing area, resulting in uneven grinding. However, when the magnetic pole rotation speed is too high, the centrifugal force on the abrasive is greater, and a few abrasives will splash out of the processing area, resulting in lower grinding efficiency and higher surface roughness values. Therefore, a larger magnetic abrasive filling quantity should be selected at high pole rotation speed to compensate for the abrasive splash problem.

### 3.6. Parameter Optimization and Validation

Data simulation optimization is performed by Design-Expect software to obtain the optimal process parameters with the minimum surface roughness value as the target according to the actual conditions of the test and the equipment. The magnetic pole rotation speed is 1200 r/min, grinding gap is 1 mm, magnetic pole feed rate is 1 mm/min, magnetic abrasive filling quantity is 2.5 g. The predicted surface roughness value Ra is 0.059 µm.

Validation tests were conducted using the processing parameters predicted by the response surface method, and the tests were repeated three times. Then observe 2D and 3D surface morphology and measure surface roughness values using a DSX1000 3D digital microscope.

The results are shown in the Table 6. The results show that the regression equations of surface roughness obtained using the response surface method on magnetic pole rotation speed, grinding gap, magnetic pole feed rate, and magnetic abrasive filling quantity have good predictive ability. The error between the actual and predicted surface roughness values obtained from the test using the optimal machining parameters is 8.5%.

Figure 13b shows the microscopic morphology map of the material surface after grinding. Comparing with Figure 13a, it can be seen that the defects such as creases and scratches on the surface of AZ31B magnesium alloy after processing are removed. The surface is flat, and the texture is more uniform. Figure 14 shows the surface roughness measured by a DSX1000 3D digital microscope for the surface morphology of the AZ31B magnesium alloy workpiece in Figure 13. By comparing the measurement results in Figure 14, the surface roughness is reduced from 0.323 μm to 0.064 μm, and a large improvement in surface quality can be seen.

## 4. Conclusions

(1)The magnetic abrasive Al_2_O_3_ prepared by plasma melting method was used for magnetic particle grinding and finishing of AZ31B magnesium alloy plate, which solved the problem of temperature rise on the surface of magnesium alloy processed by traditional method, and processed the plate with initial roughness of 0.323 μm to 0.064 μm, and improved the surface quality.(2)The force on the magnetic abrasive in the MAF process was analyzed, and the relationship between grinding pressure and magnetic induction strength was investigated, and the simulation of separate and composite magnetic pole processing was carried out by Comsol software respectively. The results show that the composite magnetic pole machining has stronger and more uniform magnetic induction strength and more obvious magnetic field gradient changes than the separate stimulation machining method, which prevents the phenomenon of MAPs fly-off or aggregation caused by insufficient magnetic induction strength during MAF, and improves the machining efficiency and surface integrity of the workpiece.(3)The regression models of magnetic pole rotation speed, grinding gap, magnetic pole feed rate and magnetic abrasive filling quantity on surface roughness were established by response surface method, and the results of residual and ANOVA proved to be a good fit. The 2D and 3D response surface plots were obtained by Design Expect software, and the influence law of different machining parameters on the surface roughness was analyzed, and the order of the influencing factors of surface roughness was obtained as: magnetic pole rotation speed > grinding gap > magnetic pole feed speed > magnetic abrasive filling quantity. The optimal process parameters obtained with the goal of obtaining the minimum surface roughness value after machining were tested, and the actual surface roughness value of 0.64 μm was obtained, with an error of 8.5% from the predicted value of 0.59 μm, which proved the validity of the model.

## Figures and Tables

**Figure 1 micromachines-13-01369-f001:**
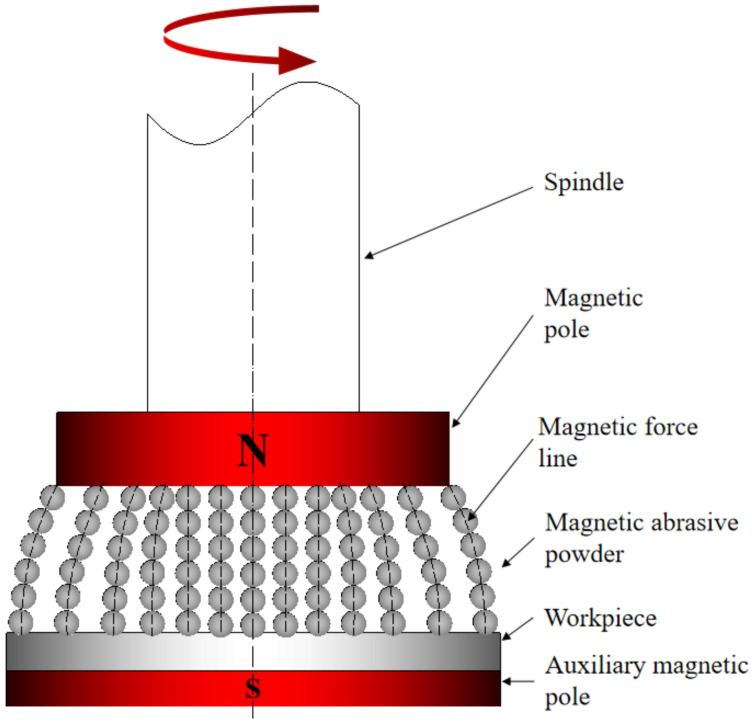
Schematic drawing of planar MAF.

**Figure 2 micromachines-13-01369-f002:**
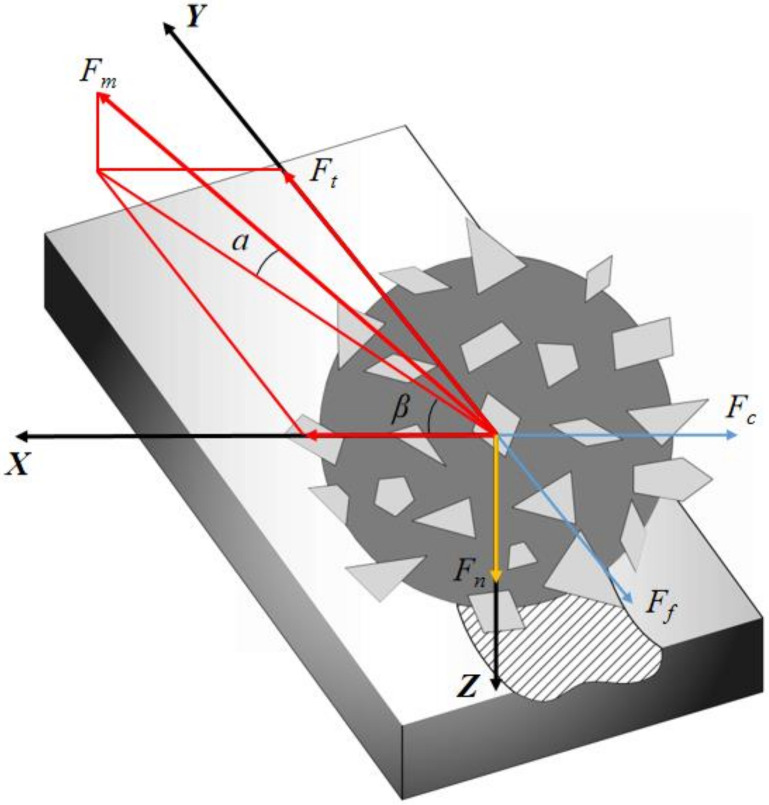
Force state of a single magnetic abrasive powder.

**Figure 3 micromachines-13-01369-f003:**
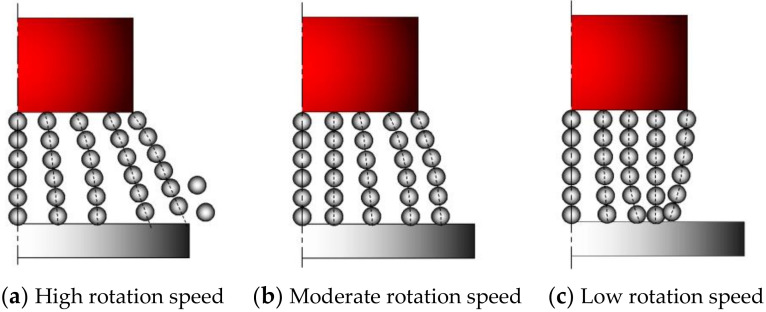
Three magnetic brush states during MAF processing.

**Figure 4 micromachines-13-01369-f004:**
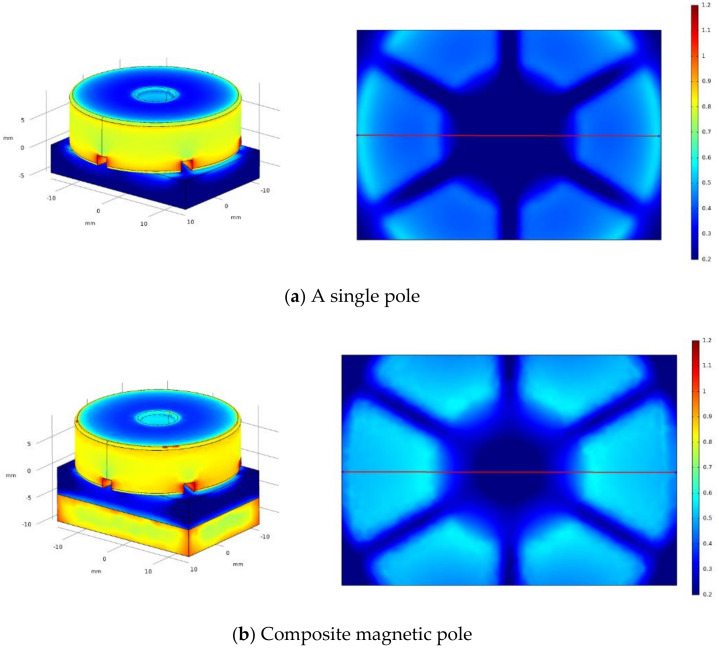
Simulation diagram of magnetic induction intensity of two MAF methods.

**Figure 5 micromachines-13-01369-f005:**
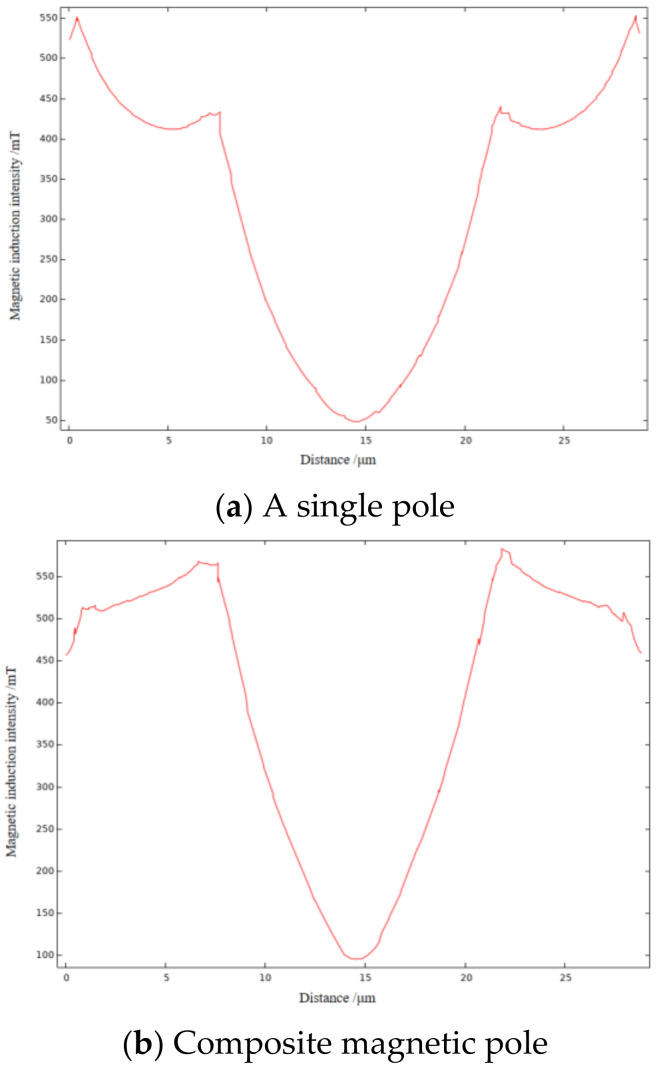
Magnetic induction intensity curve of two MAF methods.

**Figure 6 micromachines-13-01369-f006:**
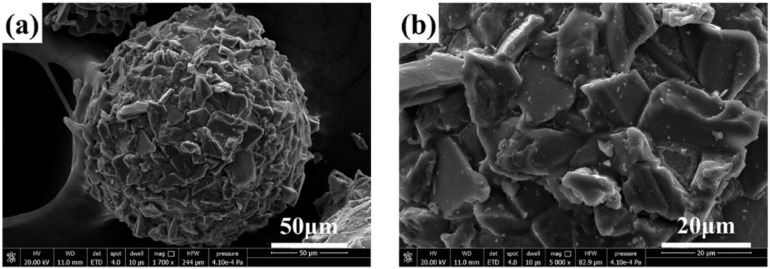
SEM micrograph of MAP: low-magnification (**a**) and high-magnification (**b**) [22].

**Figure 7 micromachines-13-01369-f007:**
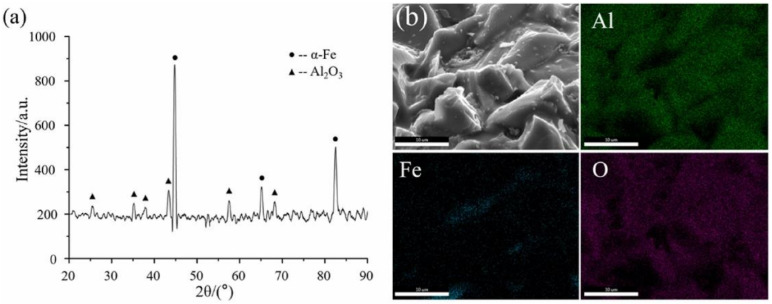
XRD graph (**a**) and EDS surface scan (**b**) of Al_2_O_3_ magnetic abrasive [22].

**Figure 8 micromachines-13-01369-f008:**
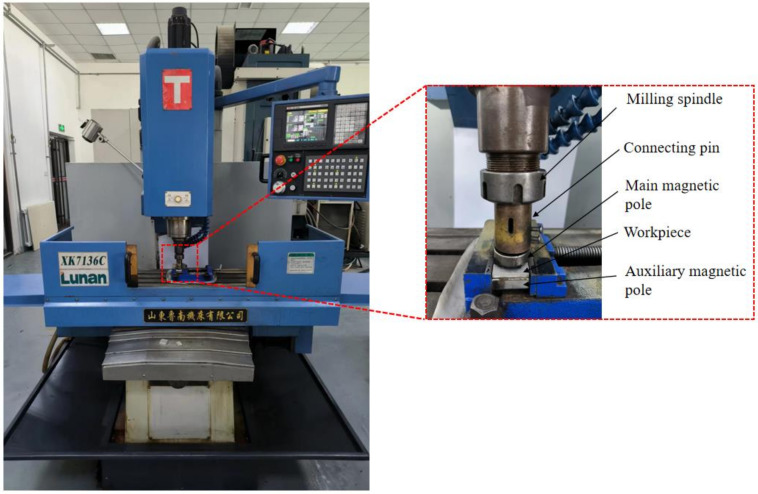
Vertical milling machine and composite magnetic pole device.

**Figure 9 micromachines-13-01369-f009:**
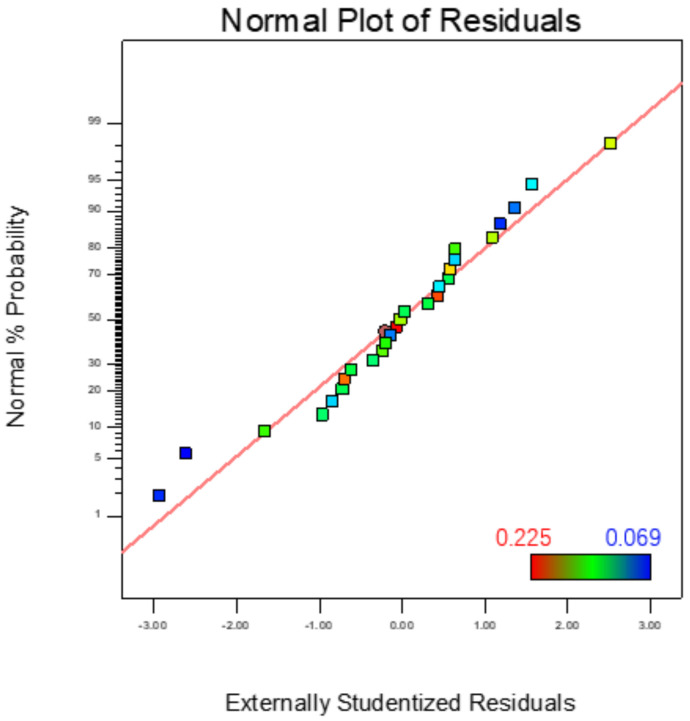
Residual normal probability distribution of surface roughness prediction model.

**Figure 10 micromachines-13-01369-f010:**
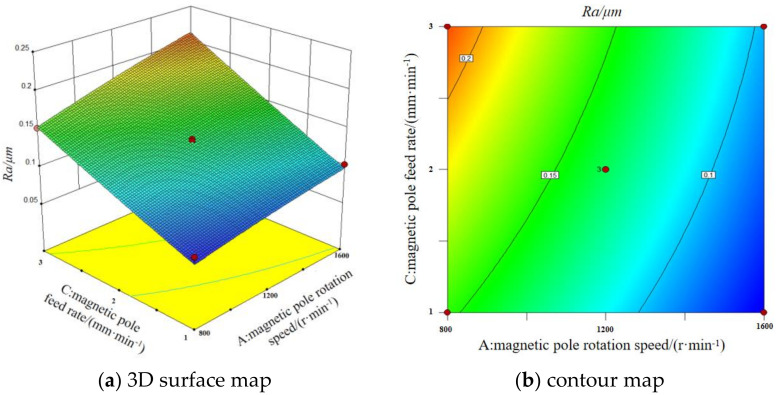
3D surface map and contour map of the interaction between magnetic pole rotation speed and feed rate on surface roughness.

**Figure 11 micromachines-13-01369-f011:**
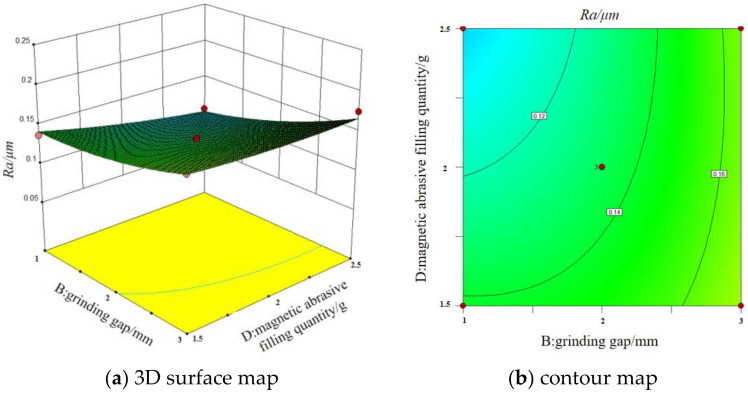
3D surface map and contour map of the interaction between grinding gap and magnetic abrasive filling quantity on surface roughness.

**Figure 12 micromachines-13-01369-f012:**
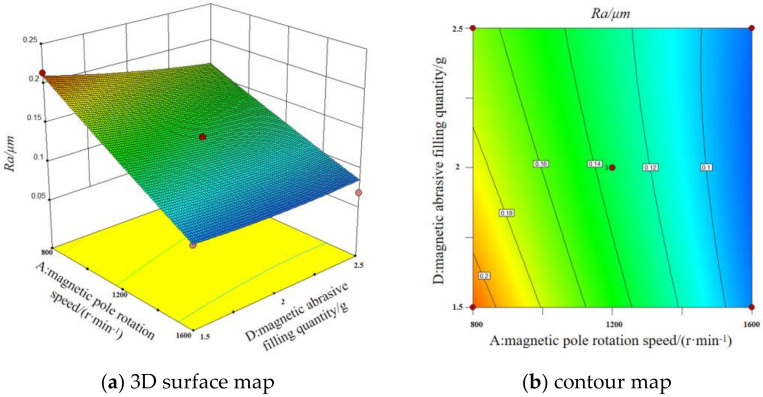
3D surface map and contour map of the interaction between magnetic rotation speed and magnetic abrasive filling quantity on surface roughness.

**Figure 13 micromachines-13-01369-f013:**
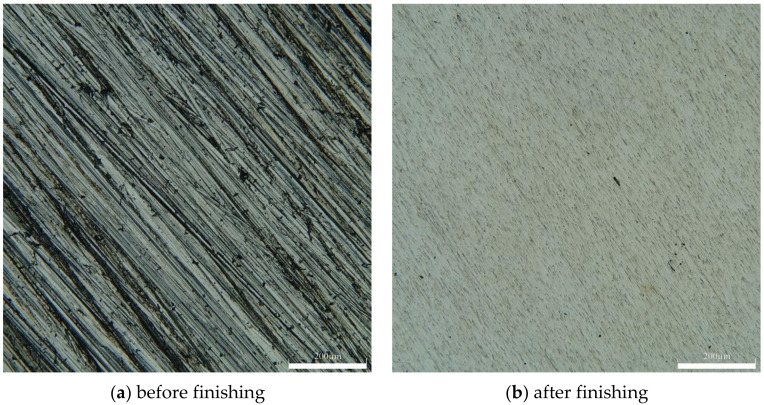
Microscopic morphology map of the material surface.

**Figure 14 micromachines-13-01369-f014:**
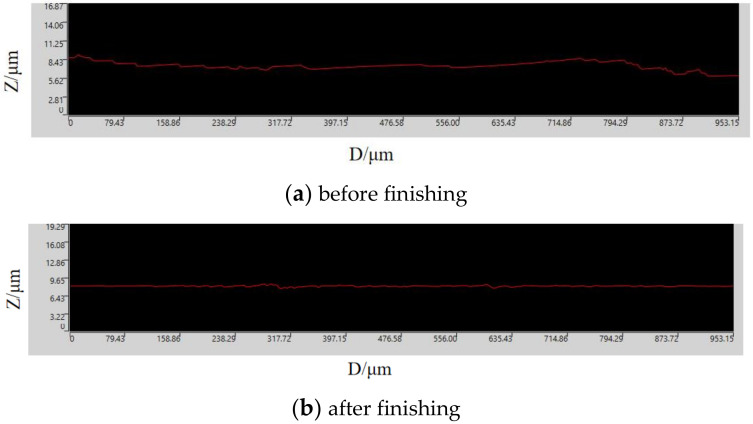
Surface roughness of AZ31B magnesium alloy workpiece.

**Table 1 micromachines-13-01369-t001:** Composition of AZ31B Magnesium Alloy [27].

Element	Al	Zn	Mn	Si	Fe	Mg
Wt.%	≤3.65	≤1.16	≤0.084	≤0.001	≤0.017	Bal.

**Table 2 micromachines-13-01369-t002:** Performance of AZ31B Magnesium Alloy [13].

Performance Indicators	Density/(g·cm^3^)	Tensile Strength (mPa)	Yield Strength (mPa)	Elastic Modulus (gPa)	Vickers Hardness (HV)
Value	1.78	290	220	45	80

**Table 3 micromachines-13-01369-t003:** Correspondence table of factor level.

Level	Magnetic Pole Rotation Speed *n*/(r·min^−1^)	Grinding Gap *d*/(mm)	Magnetic Pole Feed Rate *f*/(mm·min^−1^)	Magnetic Abrasive Filling Quantity *m*/(g)
Upper level (+1)	1600	3	3	2.5
Zero level (0)	1200	2	2	2
lower level (−1)	800	1	1	1.5
Radius of change	400	1	1	0.5

**Table 4 micromachines-13-01369-t004:** Response surface experimental dataset.

Trial	Magnetic Pole Rotation Speed *n*/(r·min^−1^)	Grinding Gap *d*/(mm)	Magnetic Pole Feed Rate *f*/(mm·min^−1^)	Magnetic Abrasive Filling Quantity *m*/(g)	*Ra*/(μm)
1	1200	1	3	2	0.131
2	1200	3	3	2.5	0.157
3	1200	2	2	2	0.137
4	1600	2	1	2	0.076
5	800	2	2	1.5	0.214
6	800	3	2	2	0.225
7	1200	3	2	2.5	0.175
8	800	2	2	2.5	0.156
9	1200	2	1	2.5	0.107
10	1200	3	3	2	0.191
11	800	2	1	2	0.152
12	1200	2	1	1.5	0.134
13	1600	2	3	2	0.102
14	1200	2	3	1.5	0.161
15	800	1	2	2	0.180
16	1600	2	2	1.5	0.088
17	1600	1	2	2	0.087
18	1200	1	2	1.5	0.137
19	1600	3	2	2	0.102
20	1200	1	1	2	0.076
21	1200	1	2	2.5	0.106
22	1200	3	1	2	0.135
23	1200	3	2	1.5	0.172
24	1200	2	2	2	0.130
25	800	2	3	2	0.208
26	1600	2	2	2.5	0.069
27	1200	2	2	2	0.134

**Table 5 micromachines-13-01369-t005:** Variance analysis of surface roughness regression model.

Source	Sum of Squares	df	Mean Square	F	Prob > F	
Model	0.047	14	3.374 × 10^−3^	22.53	<0.0001	significant
A-magnetic pole rotation speed	0.031	1	0.031	207.78	<0.0001	
B-grinding gap	6.674 × 10^−3^	1	6.674 × 10^−3^	44.57	<0.0001	
C-magnetic pole feed rate	6.075 × 10^−3^	1	6.075 × 10^−3^	40.57	<0.0001	
D-magnetic abrasive filling quantity	1.541 × 10^−3^	1	1.541 × 10^−3^	10.29	0.0075	
AB	2.250 × 10^−4^	1	2.250 × 10^−4^	1.50	0.2438	
AC	2.250 × 10^−4^	1	2.250 × 10^−4^	1.50	0.2438	
AD	3.802 × 10^−4^	1	3.802 × 10^−4^	2.54	0.1370	
BC	2.500 × 10^−7^	1	2.500 × 10^−7^	1.670 × 10^−3^	0.9681	
BD	2.890 × 10^−4^	1	2.890 × 10^−4^	1.93	0.1900	
CD	1.323 × 10^−4^	1	1.323 × 10^−4^	0.88	0.3658	
A^2^	9.481 × 10^−6^	1	9.481 × 10^−6^	0.063	0.8056	
B^2^	3.929 × 10^−4^	1	3.929 × 10^−4^	2.62	0.1312	
C^2^	2.801 × 10^−5^	1	2.801 × 10^−5^	0.19	0.6730	
D^2^	6.379 × 10^−5^	1	6.739 × 10^−5^	0.43	0.5263	
Residual	1.797 × 10^−3^	12	1.497 × 10^−4^			
Lack of Fit	1.772 × 10^−3^	10	1.772 × 10^−4^	14.37	0.0668	Not significant
Pure Error	2.467 × 10^−5^	2	1.233 × 10^−5^			
Cor Total	0.049	26				
R-Squared = 0.9634				Adj R-Squared = 0.9206		

**Table 6 micromachines-13-01369-t006:** Experimental results of optimal parameters.

Experimental 1	Experimental 2	Experimental 3	Mean Value	Predictive	Error
0.067 μm	0.065 μm	0.060 μm	0.064 μm	0.059 μm	8.5%

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
