# Peer review of "Parametric Studies on Finishing of AZ31B Magnesium Alloy with Al2O3 Magnetic Abrasives Prepared by Combining Plasma Molten Metal Powder with Sprayed Abrasive Powder"

_micromachines, 2022, doi:10.3390/mi13091369_

Round 1
Reviewer 1 Report
The paper focuses on the parametric study of finishing magnesium alloy with magnetic abrasive particles. The manuscript requires a lot of editing in the English language. Multiple citations are missing in the introductions. The modeling details are not precise and require significant rewriting. The information about measuring the experiment's response variable (surface roughness) is missing. The figures in the manuscript are not clear and precise.
Some of the following changes are highlighted below
Line 35-37 requires citation
Paragraph 74-77 requires editing. Word choices and structure of the paragraph need editing.
Figure 1: The schematic is confusing. Is the magnetic pole a monopole? Showing just the north pole of the magnet doesn't make sense. Show the magnet with both poles. Adding a schematic with the auxiliary pole will be helpful.
Section 2.2. Force analysis of MAP requires a lot of rework. Many redundant statements. Mention units for all the parameters used in the modeling. Equation numbers are missing. Mention all the assumptions for the model.
Section 2.3. Analysis of the magnetic field requires a lot of rework. References are needed for lines 177-178. Explain in detail magnetic field distribution and magnetic field gradient. Explain what is an auxiliary pole in detail. No precise results are presented in paragraphs from lines 200-209.
Figure 4: Show the base model for the magnet.
Figure 5: The graph is not clear and doesn't provide any information of additional details with respect to the experimentation.
Figure 6: EDS elemental mapping for Fig 6 (b) will be helpful
Tables 1 and 2: Are the details provided by the author or from literature? If literature values are added, cite them.
Explain how the 3D super depth of field microscope works. (line 266-267) since all the measurements were done with that.
Figure 9 (b). The Y axis has a type. Factor C is for magnetic pole feed rate.
Figure 12: Show the value of surface roughness before the finishing operation.
Reviewer 2 Report
Parametric studies on finishing of AZ31B magnesium alloy 2
with Al2O3 magnetic abrasives prepared by combining plasma 3
molten metal powder with sprayed abrasive powder
The article istitled with "Parametric studies on finishing of AZ31B magnesium alloy with Al2O3 magnetic abrasives prepared by combining plasma molten metal powder with sprayed abrasive powder" and AZ31B magnesium alloy has been grinded with Al2O3 magnetic abrasives. RSM has been applied to investigate the effect of related factors, a pridict model has been proposed and a verified experiment has been carrited out.
The work has an important practical application value. There are some mistakes in the paper.
1,"Al2O3 "in title ,absract ... should be subscript .
2,Line 29 ,"And because its elastic", "And" should be deleted.
3,Line 32"human environment,"should be "hunman body environment".
4,At the end of part 1, the research plan of this work should be added.
5,Line 75, itself is misuse.
6,Line 91, parts should be kinds/types.
7,Figure 2, Fm and Fn are not found in the figure.
8,Line 144 ... all the Fig.X should be Figure X in the paper.
9,All the equations should be listed as (1) (2)...
10,Line 166, we can say h increases with an increasing dm, but can not say they are propotional.
11,Line 194, A comsol calculation is applied ,the model conditions should be state in the paper simply.
12,Line 195-196 the sentence should be rewirte.
13,Line 201 the quality should be deleted.
14, line 294-296 A B C D and the 4 factors should be crespond with each other, such as magnetic pole rotation speed (A).
15, Line 320 have should be show.
16 ,conclusion-1,"remove the defects such as pockmarks on the surface of magnesium alloy." however, there is not related provement in the paper.
17, The ref. should be checked carefully. For example, [1] all the authors are listed ,[3] only 3 are listed, [2] the names are abbreviated form while [1] are not.
Round 2
Reviewer 1 Report
Change title for figure 7. Is it XRD of MAP? If yes, mention the XRD parameters
